# Impact of the Global Fear Index (COVID-19 Panic) on the S&P Global Indices Associated with Natural Resources, Agribusiness, Energy, Metals, and Mining: Granger Causality and Shannon and Rényi Transfer Entropy

**DOI:** 10.3390/e25020313

**Published:** 2023-02-08

**Authors:** Pedro Celso-Arellano, Victor Gualajara, Semei Coronado, Jose N. Martinez, Francisco Venegas-Martínez

**Affiliations:** 1Departamento de Métodos Cuantitativos, Centro Universitario de Ciencias Económico Administrativas, Universidad de Guadalajara, Zapopan 45100, Mexico; 2Palomar College, San Marcos, CA 92069, USA; 3Accounting, Finance and Economics Department, California State University Dominguez Hills, Carson, CA 90747, USA; 4Escuela Superior de Economía, Instituto Politécnico Nacional, Mexico City 11350, Mexico

**Keywords:** global indices, co-movement, Granger causality, DCC-GARCH

## Abstract

The Global Fear Index (GFI) is a measure of fear/panic based on the number of people infected and deaths due to COVID-19. This paper aims to examine the interconnection or interdependencies between the GFI and a set of global indexes related to the financial and economic activities associated with natural resources, raw materials, agribusiness, energy, metals, and mining, such as: the S&P Global Resource Index, the S&P Global Agribusiness Equity Index, the S&P Global Metals and Mining Index, and the S&P Global 1200 Energy Index. To this end, we first apply several common tests: Wald exponential, Wald mean, Nyblom, and Quandt Likelihood Ratio. Subsequently, we apply Granger causality using a DCC-GARCH model. Data for the global indices are daily from 3 February 2020 to 29 October 2021. The empirical results obtained show that the volatility of the GFI Granger causes the volatility of the other global indices, except for the Global Resource Index. Moreover, by considering heteroskedasticity and idiosyncratic shocks, we show that the GFI can be used to predict the co-movement of the time series of all the global indices. Additionally, we quantify the causal interdependencies between the GFI and each of the S&P global indices using Shannon and Rényi transfer entropy flow, which is comparable to Granger causality, to confirm directionality more robustly The main conclusion of this research is that financial and economic activity related to natural resources, raw materials, agribusiness, energy, metals, and mining were affected by the fear/panic caused by COVID-19 cases and deaths.

## 1. Introduction

On 11 March, the World Health Organization (WHO) declared Coronavirus, or COVID-19, a global pandemic [1]. This fact signified an unusual shock for the world, as it affected most sectors of the economy [2,3]. Shortly after the beginning of the pandemic, stock markets around the world suffered significant declines compared to those that occurred during the 2008 financial crisis, the 1987 market crash, and even the 1929 Great Depression [4]. Similarly, global commodity markets exhibited a significant drop due to supply chain disruptions that caused supply and demand mismatches [5]. For example, in March 2020, oil prices recorded their most considerable drop compared to other commodities [6].

In the financial context, the Dows Jones index fell by more than 2000 points on 3 March [7], with some sectors of the S&P 1500 index (natural gas, health care, software, among others) posting positive returns [8], while others such as the tourism, entertainment, and hospitality sectors decreased [9,10]. At the same time, the pandemic, and in particular the restrictive measures in place, had a negative impact on the management of natural resources [11] and the price of metals [12].

Different studies and investigations analyze causality, impact, co-movement, volatility, and uncertainty among economic/financial sectors, either by country, region, or a specific industry, stock market, currency, or cryptocurrency [13,14,15,16,17,18]. They have also analyzed the impact that the confirmed number of infected people or deaths had on different financial and economic activities [19,20,21,22,23]. In this regard, it is also worth mentioning the pioneering work of Baker et al. [4], who developed the Infectious Disease Equity Market Volatility Tracker Index, which includes press news from the United States regarding COVID-19, and which has been used in various investigations to measure the impact of news on the volatility of different types of financial series [6,24,25,26,27,28,29].

On the other hand, Salisu and Akanni [30] developed, in 2020, the Global Fear Index (GFI), which is a measure of fear/panic based on the number of people infected and deaths due to COVID-19. One of the relevant characteristics of the GFI lies in its coverage, since all the countries of the world are considered in its construction methodology, and consequently, for all the regions and territories of the world. GFI is daily calculated on a scale of 0–100, where zero means no fear/panic, and values closer to 100 are associated when the population feels fear/panic. GFI has been used in many applications, for instance: (1) to analyze its relationship with market volatility to determine an investment portfolio [31], (2) to measure the efficiency and coverage in the Pakistan stock market [32], and (3) to examine the influence of fear in the bond market for G7 countries [33]. In most cases, GFI has shown an important relationship with many different financial variables.

This paper examines the co-movement of GFI towards the volatility of four global indices: the S&P Global Resource Index (GRI), the S&P Global Agribusiness Equity Index (GAEI), the S&P300 Metals and Mining Index (MMI), and, finally, the S&P Global 1200 Energy Index (GEI) through the Granger causality time series using a DCC-GARCH. The DCC-GARCH model was proposed by Lu [34] and improved by Caporin and Costola [35] through simulation to obtain better confidence levels needed to accept or reject causality. It is worth mentioning that the DCC-GARCH has been widely used to analyze causality over time between pairs of economic and financial time series [29,36,37,38].

The global indices that are analyzed in this research (GRI, GAEI, MMI, and GEI) are directly and indirectly related to the primary sector of the economies. Consequently, many industries are also related to them. The interaction between the GFI (COVID-19 fear/panic) and the aforementioned global indices becomes an important issue, considering that the primary sector makes up a large part of economies. In this case, the agricultural and mining policymaker could manage its relative risk exposure in different global markets. This risk management can improve the role played by natural resources, agribusiness, energy, metals, and mining in the macroeconomic integration processes of the different sectors.

This research differs from others in that: (1) it uses a DCC-GARCH model that has several advantages for determining Granger causality; (2) it allows for identifying any immediate impact of news information on the stock market at any time, which asynchronously occurs due to how information flows [38,39]; (3) it uses dynamic cross-correlation to assess causality based on the time window width [34]; (4) it determines the causality in the mean and in a dynamic way, and, finally, (5) it determines the volatility cluster where the causality occurs [40].

This paper also demonstrates that the GFI can be considered to be a variable that could help predict the co-movement of the daily time series of all of the indices considered in this work. The results show that the GFI has a unidirectional co-movement through time with the global indices. Therefore, the GFI serves as a variable to forecast the co-movement of the rest of the variables one day in advance, which makes this anticipation relevant. Additionally, by using information-theoretic concepts, this investigation examines the causal interdependencies between the GFI and each of the S&P global indices using Shannon and Rényi directional transfer entropy flow, which is comparable to Granger causality.

This work is organized as follows: Section 2 outlines the materials and methods; Section 3 presents the results from classical time series analysis; Section 4 examines the causal interdependencies between the GFI and each of the S&P global indices using Shannon and Rényi directional transfer entropy flow, which is comparable to Granger causality; Section 5 presents a general discussion of the empirical results obtained; and, finally, Section 5 provides the conclusions.

## 2. Materials and Methods

In the following section, four global indices (the S&P Global Resource Index (GRI), the S&P Global Agribusiness Equity Index (GAEI), the S&P300 Metals and Mining Index (MMI), and the S&P Global 1200 Energy Index (GEI)) and the fear/panic index (GFI) are analyzed. The first global index is the GRI, which comprises 90 companies listed in natural resources and raw materials. Investors can diversify their investments in three sectors: agribusiness, energy and metals, and mining. The second is the GAEI, which includes 24 of the largest agribusiness companies listed on the stock exchanges around the world; investment is diversified in terms of production companies, distributors and processors, and suppliers of equipment and materials. The third is the MMI index, made up of companies that are classified in the Global Industry Classification Standard (GICS^®^). It belongs to the metals and mining sector, which produces aluminum, gold, steel, precious metals, minerals and metals, and diversified minerals. The latest global index is the GEI, which comprises energy sector companies within GICS^®^. The series were obtained from https://www.refinitiv.com (accessed on 30 December 2022).

The GFI is an index that is made up of two others: the COVID-19 cases index and the index of reported worldwide COVID-19 deaths, both with equal weights in the GFI. The series of S&P global indices are daily closing prices, and GFI is a daily index on a scale of 0 to 100, from 3 February 2020 to 29 October 2021, with a total of 425 observations. The series are transformed into logarithmic growth rates as $y_{t}=100\left(\ln\left(p_{t}\right)-\ln\left(p_{t-1}\right)\right)$. Figure 1 shows the data in nominal form and its logarithmic growth rate. It is observed that, at the beginning of studied period, the four S&P global indices fall and then they have an upward trend, declining with a valley around March 2020 and others later. The GFI index shows several changes over time, highlighting a rise at the beginning of the analyzed period with ups and downs in its trend. Regarding the rest of the variables, they present greater volatility at the beginning of the period, and at the beginning of 2021.

We present in Table 1 the descriptive statistics of the logarithmic growth rates for each time series.

All four of the S&P global series are left-skewed, and only the GFI is right-skewed. All series are platykurtic. The Jarque-Bera statistic [41] shows that the series have a non-normal distribution. To check whether the series is stationary, the RALS-LM non-parametric unit root test [42] was applied, which determines the periods of change in both the slope and the intercept. The unit root null hypothesis is rejected at 1% significance, with two periods of change. This confirms that the series lacks a unit root (see Table 1).

In order to specify the econometric model and apply the corresponding tests, we first analyzed the standardized residuals for each stationary series $\left\{y_{i,t}\right\},\;i=1,2,$ t=1,…,T, defines the sample size from a univariate GARCH(1,1) model in order to remove any autocorrelation effects. To analyze the dynamic correlation, we introduce a DCC-GARCH (1,1) model
(1)$$y_{t}\left(j\right)=\left(\begin{array}{c} y_{1,t}\\ y_{2,t-j} \end{array}\right)$$
where *j* is the lag order. As usual, the Hong test is defined as:(2)$$H_{1t}\left(k\right)=\frac{T\sum_{j=0}^{T-2}k^{2}\left(\frac{j+1}{M}\right)r_{12,t}^{2}\left(j\right)-C_{1T}\left(k\right)}{\sqrt{2D_{1T}\left(k\right)}}$$
where *M* is a positive integer and has a small impact on the size of the DCC-GARCH Hong test (we also use *M* = 2, *M* = 5, and *M* = 10, but results remain relatively constant) and *k*(*·*) is the kernel function. The other variables in Equation (2) are defined as
(3)$$C_{1T}\left(k\right)=\sum_{j=1}^{T-1}\left(1-\frac{j}{M}\right)k^{2}\left(\frac{j}{M}\right)\;$$
and
(4)$$D_{1T}\left(k\right)=\sum_{j=1}^{T-1}\left(1-\frac{j}{M}\right)\left(1-\frac{j+1}{T}\right)k^{4}\left(\frac{j}{M}\right).$$
Notice that
(5)$$H_{1,t}\left(k\right)\sim N\left(0,1\right)$$

If $H_{1,t}\left(k\right)$ is larger than the critical value of the normal distribution, then the null hypothesis of no causality is rejected. Caporin and Costola [35] mentioned that the test statistic proposed by Lu [34] must be conducted through simulations, which obtains better critical values for the null hypothesis and contrasts them with the critical values under the assumption of normality, which avoids possible type I errors.

## 3. Empirical Results from Classical Time Series Analysis

Figure 2 presents the points where the Granger causality occurs, and Table 2 shows the dates where the causality occurs. Observe now that GFI Granger causes MMI until the beginning of May 2021 in a unidirectional way. The same happens for GEI and GAEI, and there is no Granger causality for GRI. Over time, the market suffered periods of abnormal volatility due to the uncertainty generated by financial crises, political risks, or pandemics. Policies are required from governments to react in advance of the markets or to mitigate the impact to a certain extent, although the uncertainty generated by COVID-19 will continue to be present [43]. Figure 2 presents the results of $H_{1,t}\left(k\right)$. It can be observed that GFI Granger-causes MMI, GEI, and GAEI in a unidirectional way, except for GRI.

The results are in line with Ayyildiz [44], where the GFI is related to a series of agricultural products to determine their Granger causality. Furthermore, Dogan et al. [45] examined the effects of COVID-19 deaths and cases on natural resources and commodities, causing an increase in volatility [3,46,47,48]. The present research also complements the results of other studies about the relationship of COVID 19 and financial markets. For instance, Sharif et al. [47] measured COVID-19 by the number of infected cases in the U.S. Zaremba et al. [46] used government interventions, not drugs aimed at curbing the spread of COVID-19. Zhang et al. [3] based their study on global coronavirus infections obtained from the John Hopkins Coronavirus Resource Center.

Therefore, the sentiment generated by the GFI due to COVID-19 cases and deaths could affect the psychological behavior of investors. In fact, some studies have analyzed the impact of sentiment variables on stock market volatility [49,50], and others such as Jawadi et al. [51] showed that investor sentiment is one of the leading causes of asymmetric returns of the actions. Furthermore, the fear/panic caused by the combined COVID-19 cases and deaths in GFI generates a pessimistic sentiment in the market [52,53,54]. In this sense, Haroon and Rizvi [55] mentioned that the coronavirus pandemic resulted in unprecedented information coverage and outpouring of opinion in this era of rapid information, and this has created uncertainty in financial markets that leads to greater price volatility. The pandemic triggered different behaviors in different economic sectors, which, with a solid policy on the part of governments, can reduce the impact on the volatility of these series, originating a renewed economy, which brings with it an optimistic growth forecast for the coming year [43]. Compared to other public health crises that preceded this one, COVID-19 significantly impacted different markets, regardless of developed or non-developed countries [56,57].

It is worth noting that each country has a different level of COVID-19 infections and deaths, health regulations, and media exposure, which in turn affects people’s perception of fear/panic. Therefore, the perception may be different in developed and developing countries. In this sense, the perceptions of market participants in industrialized countries could have a stronger effect on global markets (or even on their own domestic markets) than market participants in emerging countries.

At this point, it is important to note that the GFI may have affected specific industries. For example, Food and Lodging Services, specifically the fast food industry, established their contactless delivery policy to provide health security for their customers. Another case was that of the restaurant sector, which began to offer its services outdoors. On the other hand, hotels offered their customers pay online or rent now and stay later services, as well as technology, to check in and checkout instead of doing it physically.

We next analyze whether the GFI serves as a variable to forecast the co-movement of the series of the S&P global indices studied. A one-day forecast is considered, applying the Granger causality test with variations in time as in [58]; which can determine local projections assuming heteroscedasticity and idiosyncratic shocks. This test allows a bivariate model to not be constrained like the recursive or mobile window models of [59], which depend on the chosen window size selection [60].

The tests are based on four statistics: the Wald exponential test (ExpW), the Wald mean (MeanW), Nyblom (Nyblom), and the Quandt Likelihood Ratio (SupLR) test, considering the Schwarz Information Criterion (SIC). An Autoregressive Vector was estimated with one lag, and a cut of 15% for the extremes. Longer lags were not statistically significant at an appropriate level. The null hypothesis is that the Wald statistics on the GFI do not cause the global indices; thus it must be rejected. Table 3 presents the results of the four tests applied to each bivariate series of the GFI considered in this study.

Figure 3 presents the sequential analysis through the time of the Wald statistic, where the Granger causality is presented one day ahead as a forecast. We observed that, among the four different tests, three were significant (except Nyblom).

The above findings demonstrate that the GFI is a variable that can help determine the co-movement of the other indices one day ahead. The GFI begins to forecast co-movement toward MMI in March 2021 until the end of the period. Regarding the GEI, GAEI, and GRI, the co-movement starts from the beginning of the period to the end, with some points where the Granger causality with heteroscedasticity and idiosyncratic shocks is not found. This shows that GFI is a variable that has a co-movement on the volatility of the indices analyzed in this study. This also indicates that the volatility of these series is sensitive to the behavior of GFI, which is based on cases and deaths from COVID-19, so the co-movement in the volatility of these indices may cause investors to not only react because of the GFI, but to the economic/financial policies that were applied during the pandemic, in order to mitigate market risk. However, false news about cases and deaths from COVID-19 should not be put aside, since they could cause an overreaction, which would generate high volatility and uncertainty in these financial markets.

The effect of cases and deaths may present a negative sentiment among economic agents, this would imply greater volatility compared to positive news. However, these agents could overreact due to the pandemic in specific periods. However, as more information arrives, the market corrects itself [61,62]. Finally, one important question is what the side effects will be on these global indices once the pandemic is over; even though different markets can be replenished, most likely it seems that uncertainty will prevail as long as the pandemic continues, and economic policies are not taken to mitigate this uncertainty.

## 4. Robustness Check with an Information-Theoretic Analysis (Shannon and Rényi Entropy)

In this section, to verify Granger causality more robustly, we follow Jizba et al. [63] by using information-theoretic concepts such as Shannon [64] and Rényi [65] information measures. We shall explore the directional information flow between the GFI and each of the S&P Global Indices. That is, we inspect Shannon and Rényi transfer entropy flow between the pairs of series. The transfer entropy flow quantifies causal interdependencies in pairs of time series. This makes the Granger causality and the Shannon and Rényi transfer entropy flow comparable.

Table 4 presents the results of the statistics of the test. Shannon’s entropy transfer results are shown in panel A and Renyi’s in panel B. Column 1 provides the direction of the information flow (→). Column 2 contains the Shannon and Rényi statistics, respectively, and column 3 is the effective entropy transfer, which was calculated using 300 shuffles. To the right of each panel are the quantiles of entropy transfer, each with its respective direction. This calculation is based on Bootstrap samples for entropy transfer estimates and non-effective transfer estimates.

Finally, it is worth noting that the empirical results show that the GFI entropy causes all the other global indices (MMI, GEI, GAEI, and GRI) with the direction of the information flow from the GFI to all the global indices, while GFI Granger causes the MMI, GEI, and GAEI in a unidirectional way, except for the GRI.

## 5. General Discussion of the Empirical Results Obtained

From the above analysis, it can be seen that GFI Granger causes the MMI, GEI, and GAEI in a unidirectional way, except for the GRI. The results are in line with Ayyildiz [44] and Dogan et al. [45]. The empirical results obtained complement the results of other studies regarding the relationship of COVID-19 and financial markets, as in Sharif et al. [47] and Zhang et al. [3].

The obtained results suggest that the feelings generated by the GFI due to COVID-19 cases and deaths affect the psychological behavior of investors, as shown in Jawadi et al. [51], who proposed that investor sentiment was one of the leading causes of asymmetric returns of the actions. In this sense, Haroon and Rizvi [55] noted that the COVID-19 pandemic produced uncertainty in financial markets that led to greater price volatility.

Another relevant finding detected was that the GFI is a variable that can help determine the co-movement of the other indices (GEI, GAEI, and GRI,) one day ahead. In other words, the GFI is a variable that has co-movement on the volatility of the indices analyzed in this study. This also indicates that the volatility of the other indices is sensitive to the behavior of the GFI.

Finally, we verified Granger causality in a more robust way by using information-theoretic concepts such as Shannon and Rényi information measures. The empirical results showed that the GFI entropy causes all the other global indices (MMI, GEI, GAEI, and GRI) with a direction of the information flow from the GFI to all the global indices, in such a way that GFI Granger causes the MMI, GEI, and GAEI in a unidirectional way, except for the GRI.

## 6. Conclusions

Our work analyzed the co-movement of the GFI, considered negative news regarding COVID-19, through Granger causality, using a DCCC-GARCH model with variations in time, during the COVID-19 pandemic period towards the volatility of the MMI, GEI, GAEI, and GRI.

The empirical results found were a unidirectional causality of the GFI towards the global indices, except for the GRI. Subsequently, we analyzed Granger causality over time with a model that included heteroscedasticity and idiosyncratic shocks to forecast a forward period of the GFI towards each of the global indices. In this case, we applied four different tests, of which three were significant. Additionally, we obtained that causality was only found from March 2021 in the GFI to the MMI. The rest of the pairs presented causality from the beginning to the end of the period. This work indicates that the GFI has a co-movement with the volatility of the other indices, and can serve as a forecast variable within these markets. Additionally, to verify Granger causality more robustly, we showed that the GFI entropy causes all the other global indices (MMI, GEI, GAEI, and GRI) with the direction of the information flow from the GFI to all of the global indices,

Finally, a limitation of this investigation could be that other economic variables should have been considered: such as the exchange rate, Gross Domestic Product, among others; and/or other indices that have emerged regarding COVID-19, such as EMV-ID, the Vaccination Index, Ciustk.cmp, among others; as well as other financial markets. These could be considered in future works. In this study, we only focused on the co-movement of the volatility of these global indices and the GFI. It should also be noted that the use of the GFI as a predictor variable has another important limitation. For example, people’s fear/panic due to COVID-19 may change as they learn to live with the virus as time passes, and therefore, people may react less to a given index value on different dates.

## Figures and Tables

**Figure 1 entropy-25-00313-f001:**
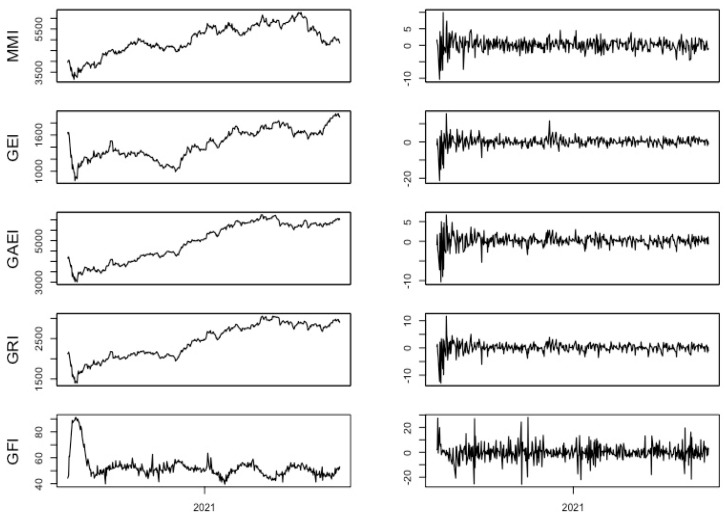
Indices and their daily logarithmic growth rates. (GFI stands for Global Fear Index, GRI stands for S&P Global Resource Index, and GAEI stands for the S&P Global Agribusiness Equity Index).

**Figure 2 entropy-25-00313-f002:**
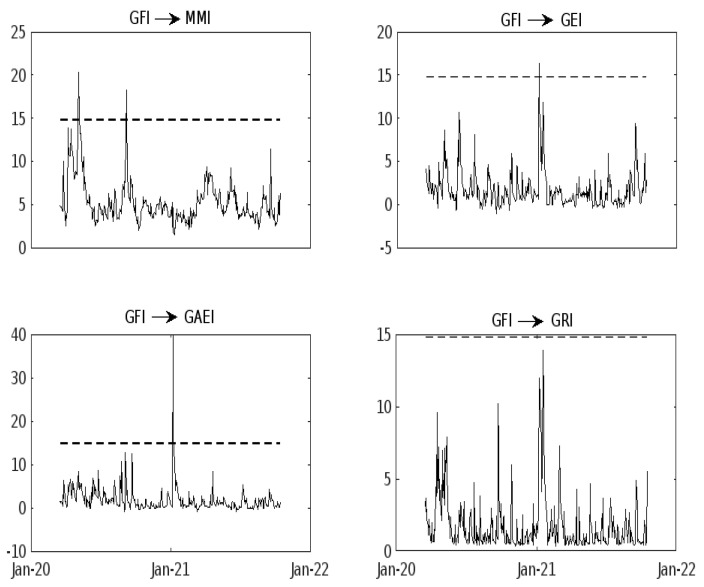
Granger causality of the DCC-GARCH test between the different indices and GFI. Note: The dotted line indicates the 99% value of the simulated critical values of the normal quantile. “→” indicates causality from one series to another. Rolling Windows were applied considering *M* = 10. The figure is our elaboration with MATLAB.

**Figure 3 entropy-25-00313-f003:**
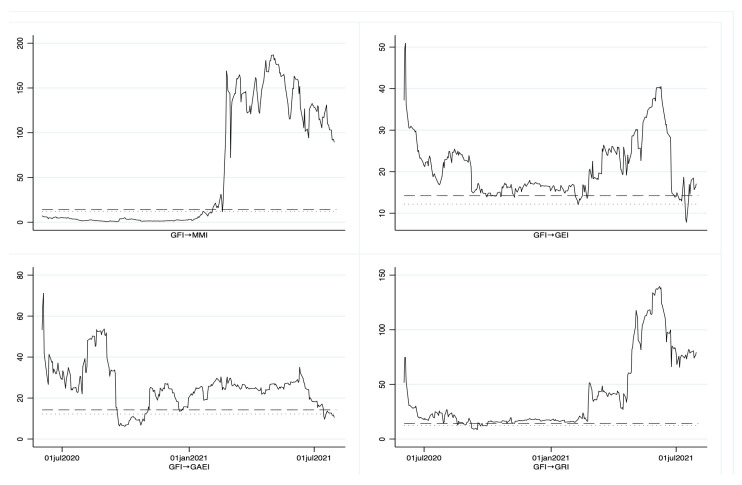
Wald statistic through time for Granger causality. Note: (---) critical level at 5%, (...) the 10% significance level, and (^__^) the Wald statistic. Calculations performed in STATA.

**Table 1 entropy-25-00313-t001:** Descriptive statistics for daily observations.

$\mathrm{Statistic}/y_{t}$	MMI	GEI	GAEI	GRI	GFI
Mean	0.02	0.00	0.06	0.05	0.04
Median	0.07	−0.04	0.10	0.14	−0.35
Min	−10.36	−21.34	−10.31	−12.91	−25.76
Max	9.95	15.52	6.69	11.69	28.06
Variance	3.71	7.69	2.46	3.56	42.9
SD	1.93	2.77	1.57	1.89	6.55
Skewness	−0.31	−1.3	−1.3	−1.41	0.18
Kurtosis	7.51	17.09	12.02	16.6	6.75
JB	369.54 ***	3681.0 ***	1683.10 ***	3667.70 ***	2018.80 ***
RALS-LM	−26.06 ***	−15.13 ***	−11.07 ***	−11.11 ***	−7.00 ***
	26 August 2021 01 August 2021	25 January 2021 23 February 2021	05 August 2020 19 August 2020	26 October 2020 25 February 2021	24 July 2020 03 December 2021

Note: (**) 5% of significance (***) 1% significance.

**Table 2 entropy-25-00313-t002:** Date of Granger Causality.

GFI → MMI	GFI → GEI	GFI → GAEI
05 April 2020 05 May 2020 05 June 2020 05 July 2020 09 July 2020	01 July 2021	01 July 2021 01 August 2021

“→” indicates causality from one series to another.

**Table 3 entropy-25-00313-t003:** Univariate Granger causality statistics.

Bivariate Series	Statistics			
	ExpW	MeanW	Nyblom	SupLR
GFI → MMI	88.53 ***	55.34 ***	2.23 *	186.94 ***
GFI → GEI	20.21 ***	21.11 ***	1.76	50.94 ***
GFI → GAEI	29.94 ***	25.53 ***	2.06	71.14 ***
GFI → GRI	65.23 ***	40.21 ***	1.90	139.60 ***

Note: (*) 10%, (**) 5% and (***) 1% level of significance.

**Table 4 entropy-25-00313-t004:** Shannon and Rényi transfer entropy flow between the GFI and each of the S&P Global Indices.

**PANEL A**								
**Shannon Transfer Entropy**			**Bootstrapped TE Quantiles**					
**Direction**	**TE**	**Eff. TE**	**Direction**	**0%**	**25%**	**50%**	**75%**	**100%**
GFI → MMI	2.5343 ***	0.06	GFI → MMI	1.88	2.03	2.07	2.11	2.23
GFI → GEI	2.2945 ***	0.06	GFI → GEI	1.75	1.87	1.91	1.95	2.11
GFI → GAEI	2.4526 ***	0.07	GFI → GAEI	1.83	1.98	2.01	2.05	2.17
GFI → GRI	2.6126 ***	0.09	GFI → GRI	1.94	2.14	2.19	2.23	2.38
**PANEL B**								
**Rényi Transfer Entropy**			**Bootstrapped TE Quantiles**					
**Direction**	**TE**	**Eff. TE**	**Direction**	**0%**	**25%**	**50%**	**75%**	**100%**
GFI → MMI	1.8436 ***	0.05	GFI → MMI	1.36	1.48	1.53	1.57	1.80
GFI → GEI	1.7925 ***	0.05	GFI → GEI	1.33	1.46	1.50	1.54	1.70
GFI → GAEI	1.921 ***	0.06	GFI → GAEI	1.43	1.52	1.56	1.61	1.74
GFI → GRI	2.0353 ***	0.07	GFI → GRI	1.46	1.68	1.74	1.80	1.98

Note: (*) 10%, (**) 5%, and (***) 1% level of significance. To calculate Shannon effective transfer entropy, the number of shuffles is 100. The bins are defined by using the 5% and 95% empirical quantiles. The weighting parameter to estimate Rényi transfer entropy is *q* = 3, which accentuates the central part of the distribution. The number of bootstrap replications for each direction of the estimated transfer entropy is 300. The number of observations that are dropped from the beginning of the bootstrapped Markov chain is 40. Lags *lx* = *ly* = 1.

## Data Availability

The test results data presented in this study are available on request from the corresponding author.

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
