# Peer review of "Impact of the Global Fear Index (COVID-19 Panic) on the S&P Global Indices Associated with Natural Resources, Agribusiness, Energy, Metals, and Mining: Granger Causality and Shannon and Rényi Transfer Entropy"

_entropy, 2023, doi:10.3390/e25020313_

Round 1

Reviewer 1 Report

This article looks at the potential effects of the Global Fear Index on various indexes of global economic activities. The authors apply several econometric tests, including a DCC_GACH Granger causality test.

The paper is timely and relevant and present insights on how people's perceptions create biases.

Some Comments.

Line 69. some words are repeated in the sentence ("the population feels").

What is the geographic scope of the GFI? is it trully global? Does GFI for specific countries area available?

Each country have different levels of Covid-19 death rates, regulations, and media exposure, which in turn affect people's perception. Also, the preceptions of market participants in developed countries would have a stronger effect in global markets (or even in their own domestic markets) than people in developing countries. 

Please specify the geographic scope of the index. 

For clarity, please define the abbreviations of figure 1 at the bottom of the figure.

Specify that the log growth rates in Table 1 are for daily observations.

Table 2. Make the table 2's col. titles more readable.

Line 243. is this a 1-day lag? Did you try with longer lags?

Please discuss the limitations of using the GFI as a predictor variable. For instance, GFI is constructed on number of cases and deads related to Covid-19. It is not a sentiment variable. People's fears due to Covid-19 may change as they learn to live with the virus, and thus may react less to the a given index value.

How many days ahead can the GFI accurately predict changes in the economic indexes? is this anticipation relevant or not? that is, if they can antcipate changes one day ahead, the predictor may not be of much value than a prediction of a month ahead.

Please discuss the effect of people's perceptions on the GFI, how they may react to said news, etc. Some industries were more affected than others (e.g. turism and leisure vs. food production). Please discuss how the index may have affected such specific industries, provide some context.

Author Response

Please find an attachment.

Reviewer 2 Report

1)             The authors build the aim as  “… to examine the effects of the GFI on a set of 19 global indexes…”  It will be better to rephrase it as the interconnection or interdependencies between GFI and indexes….

2)             In the paper is an absence of a review of the scientific background of the investigated theme

3)             From the introduction does not quite understand why it is necessary to provide such an investigation, how it will influence the improvement of the macroeconomic processes, COVID spreading, etc.  What kind of managerial decisions can be implemented after that?

4)             It is necessary to justify why such indexes can be investigated through inter-influence.

5)             After the results of the investigation should be added the discussion part

6)             The authors must check the grammar of the paper

Author Response

Please find an attachment.

Reviewer 3 Report

The article tries to detect causal (in the sense of Granger) relations between the Global Fear Index and other indices associated with natural resources and industrial productivity.

According to the article, the authors did find such causal relations in the period of the height of the pandemics.

I do not have many issuses to raise, except in what I consider to be my main area of research, that is Transfer Entropy.

1. The Shannon tansfer entropy depends on the levels of past data that is being considered (l andm m in the original formuli). The manuscript does not identify which numbers are being used.

2. The Rényi transfer entropy depends on one parameter, which determines what part of the probability distribution is is given more weight. This should be specified.

Please change (on the authors’ discretion).

“when the population feels fear/panic the population feels” to “when the population feels fear/panic”.

“confidence levels needed accept or reject” to “confidence levels needed to accept or reject”.

“made up of two other” to “made up of two others”.

“and financial markets, For instance” to “and financial markets. For instance”.

Author Response

Please find an attachment.

Round 2

Reviewer 2 Report

In general, the authors have made all necessary improvements